# A Survey of Molecular Imaging of Opioid Receptors

**DOI:** 10.3390/molecules24224190

**Published:** 2019-11-19

**Authors:** Paul Cumming, János Marton, Tuomas O. Lilius, Dag Erlend Olberg, Axel Rominger

**Affiliations:** 1Department of Nuclear Medicine, University of Bern, Inselspital, Freiburgstraße 18, 3010 Bern, Switzerland; 2School of Psychology and Counselling and IHBI, Queensland University of Technology, QLD 4059, Brisbane, Australia; 3*ABX* Advanced Biochemical Compounds, Biomedizinische Forschungsreagenzien GmbH, Heinrich-Glaeser-Strasse 10-14, D-1454 Radeberg, Germany; marton@abx.de; 4Center for Translational Neuromedicine, Faculty of Health and Medical Sciences, University of Copenhagen, 2200 Copenhagen N, Denmark; tuomas.lilius@sund.ku.dk; 5School of Pharmacy, University of Oslo, Norwegian Medical Cyclotron Centre, N-0372 Oslo, Norway and Norwegian Medical Cyclotron Centre Ltd., Sognsvannsveien 20, N-0372 Oslo, Norway; Dag.Erlend.Olberg@syklotronsenteret.no

**Keywords:** opioid receptors, positron emission tomography, radiotracers, μOR-, δOR-, κOR- and ORL1-ligands, epilepsy, movement disorders, pain, drug dependence

## Abstract

The discovery of endogenous peptide ligands for morphine binding sites occurred in parallel with the identification of three subclasses of opioid receptor (OR), traditionally designated as μ, δ, and κ, along with the more recently defined opioid-receptor-like (ORL1) receptor. Early efforts in opioid receptor radiochemistry focused on the structure of the prototype agonist ligand, morphine, although *N*-[methyl-^11^C]morphine, -codeine and -heroin did not show significant binding *in vivo*. [^11^C]Diprenorphine ([^11^C]DPN), an orvinol type, non-selective OR antagonist ligand, was among the first successful PET tracers for molecular brain imaging, but has been largely supplanted in research studies by the μ-preferring agonist [^11^C]carfentanil ([^11^C]Caf). These two tracers have the property of being displaceable by endogenous opioid peptides in living brain, thus potentially serving in a competition-binding model. Indeed, many clinical PET studies with [^11^C]DPN or [^11^C]Caf affirm the release of endogenous opioids in response to painful stimuli. Numerous other PET studies implicate μ-OR signaling in aspects of human personality and vulnerability to drug dependence, but there have been very few clinical PET studies of μORs in neurological disorders. Tracers based on naltrindole, a non-peptide antagonist of the δ-preferring endogenous opioid enkephalin, have been used in PET studies of δORs, and [^11^C]GR103545 is validated for studies of κORs. Structures such as [^11^C]NOP-1A show selective binding at ORL-1 receptors in living brain. However, there is scant documentation of δ-, κ-, or ORL1 receptors in healthy human brain or in neurological and psychiatric disorders; here, clinical PET research must catch up with recent progress in radiopharmaceutical chemistry.

## 1. Introduction

The analgesic and soporific properties of opium have been known since antiquity, perhaps first attested in the detached reveries of Homer’s Lotophagi. The sinister side of opium dreams is depicted in Tennyson’s version of that story, and more distinctly in the memoires of Thomas de Quincy, who may have had the distinction of establishing a genre of literature, the addiction diary. A key active constituent of the sap of *Papaver somniferum* was first isolated in 1804 by the apothecary Friedrich Wilhelm Sertürner, who named it morphium, later morphine (**1**). Chemists identified its elemental composition in the 19^th^ century, and efforts to determine its structure were rewarded in 1925, when Gulland and Robinson [1] recommended a structure consistent with the characteristics of morphine and codeine and their degradation products. Subsequent investigations confirmed the correctness of the analytically deduced structure of morphine, culminating in its total synthesis, achieved in the 1950s by Gates and Tschudi [2,3]. The absolute stereochemistry of morphine’s five chiral carbons (5, 6, 9, 13 and 14) was reported by Bentley and Cardwell [4] in 1955, and the first practically realizable morphine total synthesis with reasonable yields was reported by Rice in 1980 [5]. To this day, it is more economical to allow the poppy plant to do the main work of morphine (**1**) synthesis, although chemists have since produced so many structural variants that one might consider opioid pharmacology to be a discipline in its own right. There have been several reviews of opioid receptor imaging in the past decade [6,7,8], but we now present a comprehensive update on the the main classes of opioid receptor (OR) ligands used for positron emission tomography (PET), and review clinical findings with this technology. Relevant chemical structures of endogenous opioid peptides and representative small molecule opioid receptor ligands are depicted in Figure 1. 

The modern era of opioid pharmacology began with the identification of an opioid binding site in brain tissue in studies with tritiated naloxone [9]. Soon thereafter, opioid peptides were isolated from pig brain [10], which famously involved whisky as an emolument for the slaughterhouse workers. The pentapeptides Met^5^-enkephalin (**2**) and Leu^5^-enkephalin (**3**) both had morphine-like effects in inhibiting the electrically stimulated contraction of the *vas deferens*, with the latter compound being somewhat less potent. The enkephalins were most abundant in striatum and hypothalamus of rat, guinea pig and calf, and Met^5^-enkephalin (**2**) was generally 3–6 times more abundant than Leu^5^-enkephalin (**3**) [11,12]. An additional higher molecular weight opioid (β-endorphin) isolated and sequenced from camel pituitary extracts proved to be a 31 amino acid polypeptide possessing homology with Met^5^-enkephalin (**2**) [13]. A trypsin-sensitive opioid activity (dynorphin-A) isolated from pituitary is a 17 amino acid polypeptide possessing *N*-terminal homology with Leu^5^-enkephalin (**3**), with the shorter peptide dynorphin-B [1-13] having almost 1000-fold higher potency than Leu^5^-enkephalin (**3**) in the guinea pig ileum muscle preparation [14]. Soon after these discoveries, bovine DNA sequences were cloned for the β-endorphin precursor corticotropin-β-lipotropin [15], the Met^5^/Leu^5^-enkephalin (**2**, **3**) precursor preproenkephalin-A [16] (which proved to contain four copies of Met^5^-enkephalin (**2**) and one copy of Leu^5^-enkephalin (**3**), consistent with the ratio of their tissue concentrations), and the closely related preproenkephalin-B [17]. Other researchers cloned porcine preprodynorphin, the precursor for dynorphin A [1-17], dynorphin-A [1-8], dynorphin B [1-13], and other opioid peptides [18].

It was soon apparent that the endogenous opioid peptides bound to at least three distinct sites in the brain and peripheral tissues, known as μ-, δ-, and κORs. For a time, the orphaned σ receptors were thought to comprise another type of OR [20], due to the analgesic (and hallucinogenic) action of pentazocine at κORs in brain. However, binding of σOR ligands is not displacable by the opioid antagonist naloxone, nor do σORs bind opioid peptides with high affinity, such that the σ- receptor is now recognized as a pharmacological class in its own right. The μOR gene has at least 14 exons that can give rise to diverse splice variants, and at least three pharmacologically distinct subtypes are recognized: μ_1_, μ_2_ and μ_3_. Displacement studies *in vitro* and *in vivo* with the µ-selective competitive antagonist cyprodime (**6**) and the μ_1_-specific competitor naloxonazine (**7**) showed that [^11^C]carfentanil ([^11^C]Caf, **8**) binds predominantly to the μ_1_ subtype [21]. The μ_3_ subtype is alkaloid sensitive and opioid peptide insensitive; it couples to nitric oxide generation, and, extraordinarily, its endogenous agonist in amygdala seems to be morphine (**1**) [22]. There is also an opioid receptor-like receptor 1 (ORL1), which is activated by the 17 amino acid polypeptide known as nociceptin/ orphanin FQ (N/OFQ) [23].

Opioid signaling has an important function in the modulation of pain processing at the spinal level. ORs synthetized in the dorsal root ganglion are transported to peripheral nerve endings and to the superficial layers of the spinal cord dorsal horn. In the dorsal horn, µORs are the most densely expressed subtype, followed by δ-, and κORs. Over 70% of the ORs there are located on the central terminals of small-diameter (mostly C and A-delta fibres) primary afferent neurons. A main mechanism of opioid analgesia lies in the activation of presynaptic µORs in the spinal cord, leading to decreased release of excitatory transmitters and nociceptive transmission. Unforunately, PET methods do not suffice for detecting ORs in the human spinal cord.

Figure 2 shows PET images of the distributions in human telencephalon of binding sites for the four main classes of ORs, µ, δ, κ, and ORL1. The pattern of µORs in Figure 2A encompasses the telencephalic pain pathway of limbic brain regions. Supraspinal µORs in the nucleus accumbens and amygdala have a role in the analgesic and reinforcing properties of opioids. The thalamus, especially the medial structures, relay nociceptive spinothalamic input from the spinal cord to higher structures. µORs also have a prominent distribution in the brainstem, with high density in several structures associated with analgesia, such as the periaqueductal gray, rostroventral medulla, the reticular formation, and locus coeruleus [24,25,26]. From these structures, efferent outflow descends to the spinal cord where it acts to inhibit nociceptive transmission in afferent fibres. µORs are also abundant in the hypothalamus, where they might affect hormonal regulation. Receptors in the medullary vagal complex, area postrema, and nucleus tractus solitarius, can mediate endocrine actions and nausea.

As seen in Figure 2B, the δOR has high expression in the cerebral cortex, nucleus accumbens, and the caudate putamen. This receptor is involved in analgesic activity at both spinal and supraspinal sites. Similarly as µORs, agonists of central δORs contribute to respiratory depression, whereas receptors in the gut mediate constipation, an important side effect of morphine. The δORs receptors localize presynaptically where they inhibit the release of excitatory neurotransmitters [27]. Despite these properties, δ-selective drugs have not yet found clinical application.

The *κ*ORs have wide expression in rat brain, with highest levels in the ventral tegmental area, substantia nigra, nucleus accumbens, caudate putamen, claustrum, endoperiform nucleus, various hypothalamic nuclei, and the amygdala [28]. A similar expression profile occurs in the human brain [29,30], as seen in Figure 2C. Activation of *κ*ORs does not produce respiratory depression, but typical adverse effects include sedation and dysphoria, limiting the clinical use of *κ*OR targeting drugs [27]. Despite generally dysphoric effects in humans, the κOR agonist Salvinorin-A, which is obtained from the leaves of *Salvia divinorum*, finds a niche market in the drug subculture for those seeking to briefly experience a dissociative state. 

Figure 2D shows a widespread and abundant expression of NOP binding sites in human brain. Agonists of the NOP receptor, unlike µOR agonists, are devoid of reinforcing or motivational properties, but are implicated in homeostatic functions such as feeding and body weight, as well as anxiety, stress, and alcohol dependence [31].

## 2. Radiotracers for the PET Imaging of ORs

### 2.1. µOR Ligands and Non-Selective Ligands

The era of OR PET imaging was preceded by a phase of studies *ex vivo* with tritiated ligands such as the antagonist [^3^H]diprenorphine ([^3^H]DPN), which accumulated in striatum, *locus coeruleus*, *substantia nigra pars compacta*, and *substantia gelatinosa* of the living rat [32]. A similar pattern was revealed with the antagoninst [^3^H]naloxone, which showed sodium-dependent saturable binding *ex vivo*, with a B_max_ close to that seen *in vitro* [33]. In contrast, the agonist [^3^H]Foxy bound with low nM affinity at µORs *in vitro*, but failed to accumulate in brain of living rats, a property which was attributed to the presence of high sodium in the living organism. The presence of sodium in the biding medium enhanced antagonist binding *in vitro* but decreases agonist binding [34]. However, addition of Na^+^ to the incubation medium had little effect on the affinity of the morphiceptin analog µOR agonist Tyr-Pro-(*N*Me)Phe-D-Pro-NH_2_
*in vitro* ([^3^H]PL017) [35]. Unpredictable sensitivity of binding to the presence of sodium in the medium seems mainly to be a property of agonist ligands.

OR PET imaging began with the introduction of 3-*O*-acetyl-[^18^F]cyclofoxy (3-*O*-Ac-[^18^F]FcyF), (**10**). 3-*O*-Ac-[^18^F]FcyF (**10**, Figure 3) is an opioid antagonist radiotracer, which was prepared from 3-*O*-acetyl-6α-naltrexol triflate *via* direct nucleophilic substitution with tetraethylammonium [^18^F]fluoride in anhydrous acetonitrile at 80 °C for 15 min [36,37]. Based on displacement studies with CyF, binding of 3-*O*-Ac-[^18^F]FcyF (**10**) is likely to reveal the composite of µ- and κOR binding [38], despite the qualitatively µOR-like binding pattern reported in living baboon brain examined with 3-*O*-Ac-[^18^F]FcyF (**10**) [36], and the pattern of [^3^H]cyclofoxy retention in rat brain analysed *ex vivo* [39]. However, in a rat study, the increased [^3^H]cyclofoxy binding provoked by chronic treatment with morphine (**1**) could be attributed to upregulation of µOR sites [40]. Saturation binding PET studies with [^18^F]FcyF (**11**) in awake rat indicated a single binding site with apparent affinity of 2 nM and B_max_ ranging from 15 pmol/g in white matter to 74 pmol/g in striatum; these results matched closely the corresponding results obtained *in vitro* [41].

PET studies with *N*-[methyl-^11^C]-labelled morphine, codeine, heroin and pethidine indicated distinct differences in uptake and kinetics in rhesus brain [42], all seemingly in relation to lipophilicity of the various drugs. A more detailed kinetics analysis of *N*-[methyl-^11^C]pethidine (**12**) in brain of rhesus monkey indicated a very low binding potential [43]. The methadone analogue [^11^C]*L*-*α*-acetylmethadol ([^11^C]LAAM, **13**) had moderate uptake in brain of mice, but its specific binding was not reported [44]. These observations are a case in point supporting the generalization that effective pharmaceuticals do not necessary make good ligands for molecular imaging of their targets.

The displaceability of OR ligands by competitors *in vivo* is a complex matter, and one increasingly relevant given the current opioid abuse crisis in some countries. Whereas antagonists such as naloxone are effective in rescuing addicts from death by overdose, naloxone also finds experimental use in molecular imaging studies to confirm binding of PET tracers to ORs. Thus, the BP_ND_ of [^11^C]Caf (**8**) to µORs in human brain, which ranged from 1.0 in cerebellum to 2.7 in caudate nucleus, was nearly completely displaced throughout in brain by a 50-mg dose of naltrexone (NTX, **5**) [45], closely matching the dose used for resuce from opioid agonist overdose. In humans, intranasal naloxone administration caused a rapid displacement of [^11^C]Caf (**8**), in accordance with the rapid response seen in treatment for overdose [46].

Buprenorphine (BPN) [47,48,49] is a narcotic analgesic used since the 1970s in the low dose management of post-operative pain. Since 2002, BPN has approval in the United States at higher dose or in combination with naloxone (Suboxone^®^) for substitution therapy in the management of opiate addiction. BPN contains the same 6,14-ethenomorphinan skeleton as diprenorphine (DPN), and both compounds have an *N*^17^-cyclopropylmethyl substituent, although BPN contains a *tert*-butyl group in position-20 instead of methyl. Interestingly, BPN has a completely different pharmacological profile than DPN. Whereas DPN is a mixed antagonist, BPN is a partial µOR agonist and κOR antagonist, and displays some affinity for the NOP receptor [50] (Table 1).

Luthra et al. [73] synthesized *N*^17^-cyclopropyl[^11^C]methyl-buprenorphine starting from *N*^17^-*nor*-buprenorphine. Analogously to the *N*^17^-cyclopropyl[^11^C]methyl-diprenorphine synthesis [74] the corresponding precursor, *nor*-BPN, was reacted with cyclopropyl[^11^C]carbonyl chloride and the carbonyl functional group of the resulting intermediate was reduced with LiAlH_4_. Lever et al. [75] developed a metabolically stable radiotracer, 6-*O*-(methyl-^11^C)-BPN, at Johns Hopkins University in 1990 in a two-step synthesis from 3-*O*-TBDMS-6-*O*-desmethyl-BPN. The precursor was selectively alkylated in position-6 with [^11^C]iodomethane/NaH in DMF at 80 °C for two min. Following desilylation, [^11^C]BPN (**14**) was produced in 10% radiochemical yield with molar activity of 41 GBq/μmol. Subsequently, Luthra *et al.* [76], aiming to avoid the formation of 3-*O*-alkylated by-products, introduced the base-stable, acid labile trityl protecting group to protect the phenolic hydroxyl in position-3. Applying 3-*O*-trityl-6-*O*-desmethyl-BPN in a two-step, fully-automated radiosynthesis (^11^C-methylation/deprotection), yielded [^11^C]BPN (**14**) in 15% radiochemical yield and with a molar activity of 13-22 GBq/µmol. In 2014, Schoultz *et al.* [63] reported a procedure for the radiosynthesis of 6-*O*-(2-[^18^F]fluoroethyl)-6-*O*-desmethyl-BPN ([^18^F]FE-BPN (**19**)) via ^18^F-fluoro- alkylation of 3-*O*-trityl-6-*O*-desmethyl-BPN (TDBPN) precursor with [^18^F]fluoroethyl tosylate and subsequent trityl deprotection. The decay corrected formulated product yield was 26 % and the molar activity 50–300 GBq/μmol.

DPN, a semisynthetic thebaine/oripavine derivative with a methyl group amenable for labelling in position-20, belongs structurally to the ring-C bridged morphinans (6,14-ethenomorphinans, orvinols, Bentley-compounds) [48]. DPN is a nonselective OR antagonist with affinity in the nanomolar range (Table 1), 100 times more potent than nalorphine. Indeed, DPN is used in the veterinary medicine as an antidote/reversing agent/antagonist for remobilizing large African animals (rhinos/elephants, Revivon^®^), which had been immobilized with the astonishingly potent agonists etorphine or carfentanil.

The first attempt at labelling of DPN with carbon-11 in postition-20 was reported by Burns et al. [77], who used *N*-cyclopropylmethyl-dihydronororvinone as precursor. The reaction of the precursor bearing an acetyl group in position-7-alpha with [^11^C]methyllithium yielded 20-[^11^C]methyl-DPN ([^11^C]DPN (**15**)). Luthra et al. [74] developed cyclopropyl[^11^C]methyl-DPN by alkylating *N*^17^-nor-DPN with cyclopropane[^11^C]carbonyl chloride and then reducing the *N*-cyclopropyl[^11^C]carbonyl intermediate with LiAlH_4_ in THF. The radiochemical yield of the corresponding radioligands were low in both cases [74,77].

In 1987, Lever et al. [78] developed a [^11^C]DPN (**15**) synthesis by alkylating the precursor 3-*O*-TBDMS-6-*O*-desmethyl-DPN in position-6 with [^11^C]iodomethane in DMF containing sodium hydride at 80 °C for two min. After cleavage of the TBDMS protecting group, [^11^C]DPN (**15**) was obtained with 10 % radiochemical yield and 64 GBq/μmol molar activity. In 1994, Luthra and her associates at the Hammersmith Hospital developed a new precursor for the radiosynthesis of [^11^C]DPN (**15**) [76]. Selective alkylation of 3-*O*-trityl-6-*O*-desmethyl-DPN (TDDPN) with [^11^C]iodomethane in the presence of NaH/DMF (95 °C, five min). Upon deprotection with 2 M hydrochloric acid (95 °C, two min) the radiotracer **15** was obtained with a radiochemical yield of 13–19% and a molar activity of 16–24 GBq/μmol. Recently, Fairclough et al. [79] at the University of Manchester reported a modified synthetic method also starting from TDDPN, yielding [11]DPN (**15**) with ten times higher molar activity (240 GBq/μmol) [76] and a radiochemical yield of 32%. The non-selective OR partial mixed agonist/antagonist 6-*O*-(methyl-^11^C)-BPN ([^11^C]BPN, **14**) accumulated in striatum, thalamus and cingulate cortex in living baboon brain. Analysis of the dynamic PET data with a model assuming irreversible trapping gave a net blood-brain clearance (*K_i_*) of about 0.064 mL cm^−3^ min^−1^, which was halved by administration of naloxone, indicating substantial displaceability [80]. In a study in heroin addicts, the BPN occupancy at [^11^C]Caf (**8**) binding sites was estimated relative to the drug-free baseline. An oral dose of 2 mg BPN had an occupancy of about 50% throughout brain, whereas 16 mg had 85% global occupancy [81]. On the other hand, therapeutic methadone (18–90 mg/day) did not provoke any discernible occupancy at 6-*O*-(methyl-^11^C)-diprenorphine ([^11^C]DPN, **15**) binding sites, neither in human opioid addicts, nor in mice, a phenomenon attributed to high agonist potency of methadone, such that withdrawl effects are averted with a rather low occupancy [82]. In further preclinical studies from the same research group, binding of [^11^C]DPN (**15**) in mouse brain was unaltered by treatment with oxycodone (**4**) or morphine (**1**) (full agonists at µORs), but was reduced by approximately 90% by BPN (partial agonist at µORs and antagonist at the δ- and κORs).

A comparative OR PET study in humans compared the distributions of the µOR-selective agonist [^11^C]Caf (**8**) and the mixed antagonist [^11^C]DPN (**15**) [83]. Qualitatively, [^11^C]DPN (**15**) binding in the striatum, cingulate and frontal cortex exceeded that of [^11^C]Caf (**8**) (which had highest binding in the µOR-rich thalamus), consistent with labeling of additional non-µOR sites by [^11^C]DPN (**15**). An investigation of ORs in human cerebellum showed abundant binding of a µOR-specific ligand in the molecular layer, moderate binding of a κOR-selective ligand, but a near absence of δOR binding sites, which was consistent with the observations of [^11^C]DPN (**15**) binding *in vivo* [84]. The presence of binding sites in cerebellum can be an obstacle to the valid use of reference tissue methods of PET quantitation.

As a longer-lived alternative to [^11^C]DPN, (**15**), Wester et al. developed 6-*O*-(2-[^18^F]fluoroethyl)-6-*O*-desmethyl-DPN ([^18^F]FE-DPN (**16**)) [85], which contains a 2-fluoroethoxy group in position-6 instead of an OCH_3_. [^18^F]FE-DPN (**16**) was synthesized from TDDPN, the so called *“Luthra-precursor”* [76], the same precursor as for [^11^C]DPN (**15**). For the synthesis of **16**, TDDPN was reacted with [^18^F]fluoroethyl tosylate ([^18^F]FE-Tos) in DMF in the presence of sodium hydride for five min at 100 °C. The trityl protecting group was removed with 2 N hydrochloric acid, yielding [^18^F]FE-DPN (**16**) with a radiochemical yield of 22 ± 7% and the molar activity was 37 GBq/μmol [85]. In 2013, Schoultz *et al.* [86] reported an automated radiosynthesis of **16** from TDDPN with a decay-corrected radiochemical yield of 25 ± 7%. [^18^F]FE-DPN (**16**) has similar uptake as [^11^C]DPN (**15**) in mouse brain, and obtained a BP_ND_ in human thalamus of about 2 relative to occipital cortex, versus only 0.3 in somatosensory cortex [87]. Women showed faster plasma metabolism [^18^F]FE-DPN (**16**) than men, which might contribute to apparent gender differences in binding [88].

The first instance of full compartmental analysis of an opioid PET ligand in living brain was for the case of [^11^C]Caf, as described below. This fentanyl analogue belongs to the 4-anilidopiperidine (4AP) class of OR ligands, which are potent µOR-selective agonists. Since 1960, numerous 4AP-type OR ligands were synthesized and their structure-activity-relationship at ORs were recently summarized [89,90]. Caf is structurally different from fentanyl in that it contains an additional carboxymethyl group in position-4 of the piperidine ring. Caf is a µOR-selective full agonist of extreme potency, being almost 10,000 times more potent than morphine (**1**) [58,91]. In 1985, [^11^C]Caf **(8)** radiotracer was applied in the first human PET study [59,92]. For the radiosynthesis of [^11^C]Caf, desmethyl-Caf sodium carboxylate was alkylated with [^11^C]iodomethane in DMF at 35 °C for five min [92]. This procedure gave molar activity at the end of synthesis of 122 GBq/µmol, which would correspond to mass dose of about 500 pg in a human PET study, which is too low to have any effect on particpants. According to a novel version of the radiosynthesis, desmethyl-Caf free acid serves as precursor in a reaction performed in dimethylsulfoxide with [^11^C]methyl triflate in the presence of tetrabutylammonium hydroxide [93]. This procedure gave a molar activity of 5 GBq/µmol, which would correspond to a mass dose of 100 µg, certainly intruding into the range causing some pharmacological effects. Risk of toxicity is a serious matter in PET imaging with potent agonists, and for society in general, given the weaponization [94] (figurative and literal) that is possible with Caf.

For the compartmental analysis of [^11^C]Caf (**8**), two models were fitted to dynamic time activity curves (TACs) measured by PET in human brain relative to a metabolite-corrected arterial input function [95]; this approach is the gold standard for PET quantitation. The input function obtained by HPLC analysis of plasma extracts showed that untransformed parent fractions declined to 50% at 25 min post injection. The authors estimated microparameters for the reversible transfer of the tracer across the blood-brain barrier (K_1_/k_2_), the reversible binding to a receptor compartment (k_3_/k_4_), and the reversible association to a non-specific compartment in brain (k_5_/k_6_). The mean binding potential (BP; k_3_/k_4_) was 1.8 in frontal cortex and 3.4 in thalamus at baseline, versus only 0.16 and 0.26 after treatment with naloxone (0.1 mg/kg). This study set a very high standard for quantitative PET analysis, although the molar activity of the tracer may not have been completely adequate. The amount of mass injected corresponded to about 5 µg Caf per scan, which could cause some analgesia in humans, although being less than 10% of the dose causing loss of consciousness. Nonetheless, this again raises the issue of safety in PET studies with the using of high-potency full agonist ligands, as noted above. Unless the the highest possible molar activity is obtained, pharmacologically significant occupancy can occur, bringing a risk of toxicity. Test-retest studies with [^11^C]Caf (**8**) showed admirable low variability (< 6%) and high intraclass correlation coefficients (ICC > 0.90) of the total distribution volume (V_T_) relative to the metabolite-corrected arterial input, and likewise BP_ND_ relative to a reference tissue [96].

Phenethyl-orvinol (PEO) [97] shares the same in ring-C bridged morphinan scaffold as DPN and BPN. PEO contains a 6,14-*etheno*-bridge, an *N*^17^-methyl substituent and a 2-phenethyl group in position-20. It is a full agonist with higher affinity at µOR (0.18 nM) and κORs (0.12 nM) than to δORs (5.1 nM). The radiosynthesis of 6-*O*-(methyl-^11^C)-phenethyl-orvinol ([^11^C]PEO, **18**) was reported by Marton *et al*. [62] in 2009. The *Luthra-type* trityl-protected precursor (3-*O*-trityl-6-*O*-desmethyl-phenethyl-orvinol, TDPEO) was alkylated in position-6 with [^11^C]iodomethane in the presence of 8-10 equiv. sodium hydride. The protecting group was removed with 1 M hydrochloric acid in ethanol, yielding [^11^C]PEO (**18**) with a radiochemical yield of 57 ± 16% and a molar activity of 60 GBq/μmol.

In 2012, Marton and Henriksen [98] reported the preliminary results of the synthesis of 6-*O*-(2-[^18^F]fluoroethyl)-6-*O*-desmethyl-phenethyl-orvinol ([^18^F]FE-PEO, **17**) starting from 6-*O*-(2-tosyloxyethyl)-6-*O*-desmethyl-phenylethyl-orvinol (TE-TDPEO) *via* direct nucleophilic fluorination and subsequent deprotection. This procedure gave [^18^F]FE-PEO (**17**) in an isolated preparative yield of 35 ± 8% with a molar activity of 55–130 GBq/μmol. In 2013, a research group of the University of Cambridge [99] investigated [^18^F]FE-PEO (**17**) as a candidate OR PET-ligand, obtained by an automated cGMP-compliant method the [^18^F]FE-PEO at 28 ± 15% yield and 52–224 GBq/μmol molar activity. In 2014, Schoultz et al. [63] reported the synthesis and biological evaluation of three structurally-related 6-*O*-(2-[^18^F]fluoroethyl)-6-*O*-desmethy-orvinols, i.e. [^18^F]FE-DPN (**16**), [^18^F]FE-BPN (**19**), and [^18^F]FE-PEO (**17**). The production of these ^18^F-fluoroethyl-orvinol radiotracers (**16**,**17**,**19**) was accomplished from 3-*O*-trityl-6-*O*-desmethyl-orvinol precursors (TDDPN, TDBPN, TDPEO) in a two-pot, three-step synthesis. [^18^F]FE-PEO (**17**) had a molar activity at end of synthesis of 50–250 GBq/μmol [99], corresponding to an injected mass <1 µg in human PET studies. The total distribution volume (V_T_) in rat brain ranged from 1 mL cm^−3^ in cerebellum to 8 mL cm^−3^ in thalamus; displacement studies *in vitro* with the µOR-selective agonist DAMGO indicated high specificity in certain brain regions. [^18^F]FE-DPN (**16**) had a molar activity of 37 GBq/μmol [85].

### 2.2. Delta Ligands

*N*1′-Methylnaltrindole (MeNTI, Figure 4) is a highly selective δOR antagonist (Table 1). MeNTI was prepared from naltrexone (**5**) in a Fischer-indol synthesis with *N*-methyl-*N*-phenylhydrazine [64]. The radiosynthesis of [^11^C]MeNTI was reported by Lever *et al.* in 1995 [100]. In the first step, 3-*O*-benzyl-naltrindole was reacted with [^11^C]iodomethane in DMF in the presence of either sodium hydride or tetrabutylammonium hydroxide at 80 °C for two min. The next step was hydrogenolysis of the formed 3-*O*-benzyl-*N*1′-(methyl-^11^C)-naltrindole under heterogenous catalytic conditions (H_2_, 10% Pd/C, DMF/ethanol, 80 °C, four min), or alternatively catalytic transfer hydrogenation (HCOONH_4_, 10% Pd/C, MeOH). This gave [^11^C]MeNTI (**22**) with 6% radiochemical yield and a molar activity of 76 GBq/μmol.

Human PET studies with [^11^C]MeNTI (**22**) confirmed earlier demonstrations in living mice of δOR-selectivity *in vivo* [101]. The binding ratio relative to cerebellum ranged from 1.1 in hippocampus to 1.7 in striatum and insular cortex, regional values correlated rather precisely with known density of δORs *in vitro*, and showed 50% displacement after administration of 50 mg NTX (**5**). The tracer showed pseudo-irreversible binding characteristics over the course of 90 min, with net blood-brain clearance (*K_i_*) ranging from 0.04 in cerebellum to 0.11 mL cm^−3^ min^−1^ in putamen [102]. The *K_i_* for [^11^C]MeNTI (**22**) in human brain declined by only about 20% after treatment with naloxone at a dose completely displacing µOR sites [45]. The authors of that study suggested that incomplete and variable δOR blockade might contribute to the success of NTX (**5**) as a treatment for alcoholism.

*N*1′-(2-[^18^F]fluoroethyl)naltrindole (**23**, [^18^F]FE-NTI, BU97001) was developed by Matthews et al. in 1999 [103]. The precursor, *N*1′-[2-(tosyloxyethyl)]-3-*O*-benzyl-naltrindole, was prepared in four consecutive transformations from naltrexone (**5**). In the first step, naltrexone (**5**) was reacted in a Fischer-indol synthesis with 2-(N*1*-phenylhydrazino)acetic acid ethyl ester. The resulting indolomorphinanyl-acetic ester was reacted with benzyl bromide to yield the 3-*O*-benzyl protected NTI derivative, which was reduced with LiAlH_4_ in THF-toluene to afford the corresponding indolomorphinanyl ethanol intermediate. This compound was reacted with tosyl chloride to provide the appropriate precursor with a tosyloxy leaving group. For the radiosynthesis of [^18^F]FE-NTI (**23**), the precursor was reacted in a direct nucleophilic reaction with potassium [^18^F]fluoride/K_2_CO_3_/Kryptofix[2.2.2] in DMF to yield *N*1′-(2-[^18^F]fluoroethyl)-3-*O*-benzyl-naltrindole. Final debenzylation by hydrogenolysis under heterogenous catalytical conditions (H_2_, Pd/C, *N*,*N*-dimethyl formamide) gave [^18^F]FE-NTI (**23**) with a radiochemical yield of 10% and molar activity of 31 GBq/µmol. [^18^F]FE-NTI (**23**) was an antagonist in mouse *vas deferens* with high selectivity over μ- and κOR sites, and its tritiated version bound to rat whole brain as a single site with K_D_ of 0.42 nM and B_max_ of 3 pmol g^−1^ [104].

In 2007, Bourdier et al., [105] reported the radiosynthesis of a 2-[^11^C]methyl- pyrrolo[3,4-b]pyridine-5,7-dione derivative (*N*-substituted-[^11^C]quinolinimide) (**24**). The radiotracer containing a [^11^C]methyl-group on the pyridine ring was synthesized from a tributylstannyl precursor, with introduction of the [^11^C]methyl group by the Stille reaction using [^11^C]iodomethane in the presence of *tris*(dibenzylideneacetone)dipalladium, tri-*o*-tolylphosphine, K_2_CO_3_, and CuCl in DMF, heated at 90 °C for five min. The labelled compound (**24**) was synthesized with a radiochemical yield of 60 ± 10% and a molar activity of 30–56 GBq/μmol. The unlabelled version of the *N*-substituted quinolinimide had higher δOR-selectivity than MeNTI, but its ^11^C-derivative (**24**) failed to label ORs in mouse brain, due either to excessively rapid metabolism [105], or its only moderate affinity.

### 2.3. Kappa Ligands

GR89696 ((±)-methyl 4[(3,4-dichlorophenyl)acetyl]-3-(1-pyrrolidinylmethyl)- 1-piperazine- carboxylate [106,107,108] (Glaxo Group Research Ltd.) is a κOR-selective agonist with an arylacetamidopiperazine/diacylpiperazine structural core. GR103545, the biologically active (*R*)-(−)-enantiomer of GR89696, displays subnanomolar affinity and 1000-fold selectivity for human κOR (*K_i_* = 0.02 nM), [65]. Ravert et al. [109] reported synthesis of both enantiomers of [^11^C]GR89696 (**26**, Figure 5) from the corresponding chiral normethylcarbamoyl precursor [108,110]. The radiosynthesis was accomplished by acylation of the secondary amine with [^11^C]methyl chloroformate in dichloromethane in the presence of trimethylamine, giving product with molar radioactivity of 69 GBq/μmol. Mouse brain distribution of the synthesized enantiomers, ((*R*)-(−)-[^11^C]GR103545 (**25**) and the (S)-(+)-enantiomer [^11^C]SGR) was determined *in vivo*, which showed the (*S*)-enantiomer to be inactive. The low radiochemical yield of the radiosynthesis (2–14%) [110,111] motivated the development of elaborate new radiochemical methods. In 2008, Schoultz et al. [112] developed a simple [^11^C]methyl triflate mediated methylation of carbamino adducts. Normethylcarbamoyl-GR103545 was converted to [^11^C]GR103545 (**25**) with [^11^C]CH_3_OTf under mild conditions in 64–91% radiochemical yield. Wilson et al. [113] developed a method for preparing [^11^C-carbonyl]-methylcarbamates directly from primary or secondary amines, applying either DBU or BEMP and cyclotron-produced [^11^C]CO_2_. [^11^C-carbonyl]-GR103545 (**25**) was synthesized with high radiochemical purity (>98%) and molar activity of 108–162 GBq/μmol. In 2011, Nabulsi et al. [114] reported an automated two-step, one-pot procedure for the synthesis of [^11^C]GR103545 (**25**) from normethylcarbamoyl-GR103545 *via* transcarboxylation using the zwitterionic carbamic complex DBU-CO_2_ and [^11^C]CH_3_OTf. 

In PET studies, the κOR-agonist ligand [^11^C]GR103545 (**25**) had a V_T_ in baboon brain ranging from 3 mL cm^−3^ in cerebellum to 10 mL cm^−3^ in striatum and cingulate cortex [111]. Naloxone homogeneously displaced tracer throughout brain, but had no effect on V_T_ in cerebellum, which would support use of that brain region as a reference tissue. The tracer had >100-fold selectivity for κ- over δ- and µORs *in vitro* [65]. Much as in baboons, PET studies in rhesus monkey showed V_T_ ranging from 8 mL cm^−3^ in cerebellum to 21 mL cm^−3^ in striatum. Other monkey studies showed BP_ND_ ranging from 0.3 in amygdala to 2.2 in putamen [115]. This bolus plus infusion study with increasing mass dose in monkeys indicated an *in vivo* K_D_ of 2 nM and B_max_ of 1–6 pmol g^−1^. In 2014, Naganawa et al. [116] reported the first-in-human PET study with [^11^C]GR103545 (**25**). Test-retest variability of the quantitative endpoint V_T_ was about 15%, and binding ranged from 8 mL cm^−3^ in cerebellum to 41 mL cm^−3^ in amygdala; Lassen plots with naltrexone blocking indicated a non-specific uptake (V_ND_) of only 3.4 mL cm^−3^, thus emphasizing the absence of any non-binding reference region.

Based on the substituted-diacylpiperazine scaffold of GR103545, researchers at Yale University developed the new κOR agonist radiotracers [^11^C]EKAP (**27**) [68] and [^11^C]FEKAP (**28**) [67] with improved pharmacological and PET-imaging profile compared with the native compound. In the open-ring analogs of GR103545, the pyrrolidinyl-methyl group of the original molecule in position-3 was replaced by a diethylamino-methyl in EKAP and a ((ethyl)2-fluoroethyl)amino)methyl group in FEKAP. Imaging studies [^11^C]EKAP (**27**) in rhesus monkey showed rapid metabolism *in vivo* and fast, reversible binding kinetics in brain that was blockable with specific competitors. The BP_ND_ ranged from 0.8 in frontal cortex to 1.8 in globus pallidus.

Researchers at Eli Lilly, in cooperation with the Yale University, developed κOR antagonist radiotracers with the 3-pyridinyl-1-pyrrolidinylmethyl structural scaffold. Along these lines, in 2013, Zheng et al. [57] synthesized the selective κOR antagonist radiotracer [^11^C]LY2795050 (**30**) from an iodophenyl precursor in a two-step procedure. The precursor was converted by transition metal-mediated cyanation using H^11^CN and Pd_2_(dba)_3_/dppf to a [^11^C]nitrile intermediate. This latter was partially hydrolysed with NaOH/H_2_O_2_ in DMF at 80 °C for five min, giving a 12% radiochemical yield with molar activity of 23.6 GBq/μmol. [^11^C]LY2459989 (**31**) was prepared in a two-step one-pot radiosynthesis. In the first step, an aryl-iodide-type precursor was transformed in a palladium catalyzed reaction (Pd_2_dba_3_/dppf) with H^11^CN to the corresponding [^11^C]nitrile, which was reacted with H_2_O_2_ under basic condition to afford **31** with 7.4% radiochemical yield and 23 GBq/μmol molar activity.

LY2459989 is the fluorine-containing analogue of LY2795050. Li *et al.* [117] synthesised the ^18^F-fluorine-labelled version of LY2459989 using two different methods. Using the nitro precursor, the radiochemical yield was too low, but applying the iodonium ylide precursor, [^18^F]LY2459989 (**30**) was prepared with 36% radiochemical yield and 1,175 GBq/μmol molar activity. While admirably high, this molar activity falls far short of the theoretical maximium for ^18^F-, which is 63,000 GBq/µmol. Where does all that non-radioactive fluoride come from?

As noted above, salvinorin A [118,119,120] is a naturally occurring non-alkaloid neo-clerodane diterpenoid, isolated from *Salvia divinorum*. Also as noted above, smokings Salvinorin-A can provoke a dissociative hallucinogenic experience distinct from that of the classical hallucinogens. It has a unique structure with seven chiral carbons and is potent and highly selective κOR agonist; salvinorin A does not display any significant activity at other OR subtypes. In 2008, Hooker *et al.* [121] synthesized the carbon-11 labelled version of salvinorin A (**33**), using salvinorin B as precursor for the radiosynthesis. The 2-alpha-hydroxyl group of the precursor was acylated with [^11^C]acetyl chloride in DMF in the presence of DMAP at 0 °C for 7–10 min, giving a radiochemical yield of 3.5–10% with molar activity 7.4–28 GBq/μmol. PET studies in baboon brain showed rapid uptake and washout of [^11^C]salvinorin A (**33**), matching the brief duration of the hallucinatory/dissociative experience reported by humans users. Rat studies showed that acute doses of salvinorin A caused dose-dependent occupancy at brain κORs labelled *in vivo* with [^11^C]GR103545 (**25**). Pretreatment with a high dose in the hours before PET examination caused persistent reductions in receptor availability, despite the brief plasma half-life of the drug, and despite the rather brief duration of the hallucinogenic experience. This suggests that κOR activation by agonists such as salvinorin A provokes a delayed and persistent receptor internalization [122].

In 2001, Thomas et al. [123] identified the first κOR-selective antagonist ligand, *trans*-3,4-dimethyl-4-(3-hydroxyphenyl)piperidine (JDTic, > 200-fold selective, Table 1), with non-opiate structure. MeJDTic is a derivative of JDTic that is ring methylated on the nitrogen of the tetrahydroisoquinoline. Poisnel et al. [69] prepared [^11^C]MeJDTic (**29**) from JDTic by methylation with [^11^C]methyl triflate in acetonitrile at room temperature for three min. The radiochemical yield was 78–98% and the molar activity 1.5–4.4 GBq/μmol. Recently, Schmitt et al. [124] synthesized *N*-[^18^F]fluoropropyl-JDTic ([^18^F]FP-JDTic) (**35**)) from JDTic with [^18^F]fluoropropyl-tosylate in DMSO in the presence of DIPEA and LiI at 150 °C for 30 min. *In vivo* studies in mouse showed accumulation of [^18^F]FP-JDTic (**35**) in peripheral organs rich in κORs. [^11^C]MeJDTic (**29**) entered mouse brain *in vivo*, albeit attaining a concentration of only 0.2–0.3% ID/g. Its binding was substantially reduced by treatment with the κOR agonist U50,488, but was unaffected by morphine (**1**) or naltrindole (NTI), thus attesting to its high selectivity for κOR sites [69]. The κOR-agonist [^11^C]LY2795050 (**30**) had an *in vitro* binding affinity of 1 nM, and 25 fold selectivity over µORs (Table 1). It readily entered monkey brain, and was substantially displacement by naloxone [57]. Displacement studies showed scant specific binding in monkey cerebellum, which supports its use as a reference region for quantitation. The tracer had a BP_ND_ as high as 1.0 in parts of the basal ganglia [125]. Dual tracer studies in monkey showed the LY2795050 displaced [^11^C]Caf (**8**) from µOR sites with an ED_50_ of 119 μg/kg, whereas the ED_50_ at κOR sites was 16 μg/kg, indicating 8-fold selectivity *in vivo* [125,126]. Corresponding human studies with kinetic modelling showed V_T_ ranging from 2 mL cm^−3^ in cerebellum to 4 mL cm^−3^ in amygdala, and Lasson plots with partial NTX (**5**) blocking indicated a V_ND_ (non-specific binding) close to 1.6 mL cm^−3^, thus giving a BP_ND_ of 1.5 in amygdala versus only 0.2 in cerebellum [126]. The test-retest reliability in human brain was about 10% [127]. PET with [^11^C]LY2795050 (**30**) has revealed the dose-occupancy relationship in human brain for the experimental κOR antagonist LY2456302 (**34**), which is under development as a treatment of alcoholism [128].

[^11^C]LY2459989 (**31**) had sub-nM affinity at κOR sites *in vitro*, with 30-fold selectivity over µOR and 400-fold selectivity over δOR sites [66]. Preliminary PET studies in monkey showed rapid kinetics and substantial displaceability *in vivo*, with BP_ND_ ranging from 0.5 in thalamus to 2.2 in globus pallidus. A comparison of κOR ligands in rat showed that the agonist [^11^C]GR103545 (**25**) and the antagonist [^11^C]LY2459989 (**31**) had similar displacement by various κOR antagonists. However, and of great significance, the κOR agonists salvinorin A and U-50488, while displacing [^11^C]GR103545 (**25**) binding *in vivo*, did not alter [^11^C]LY2459989 (**31**) binding [129], which may indicate an allosteric binding mechanism. The novel κOR agonist tracer [^11^C]-EKAP (**27**) showed fast uptake kinetics and high specific binding in monkey brain, with V_T_ ranging from 12 mL cm^−3^ in cerebellum to mL cm^−3^ in globus pallidus, corresponding to a BP_ND_ of 1.8, its binding was 95% displaced by pre-blocking with the antagonists naloxone or LY2795050 [68].

The highly selective and potent κOR-ligand U-50488 served as a scaffold for developing fluoro-alkylated PET ligands, but proved inappropriate due to 100-fold loss of affinity relative to the starting compound [130]. The novel fluorinated κ-ligand [^18^F]LY2459989 (**32**) had similar kinetic properties in monkey PET studies to those of [^11^C]LY2459989 (**31**) [117].

### 2.4. Nociceptin and Opioid-like 1 Receptors (ORL1)

Emerging evidence supports the use of agonists for the nociceptin/orphanin FQ peptide receptor (NOP) in the clinical management of pain and for substance abuse [131], thus presenting an attractive target for molecular imgaing A series of NOP ligands based on a 2′-fluoro-4′,5′-dihydrospiro[piperidine-4,7′-thieno[2,3-c]pyran]-scaffold were screened in rats [71]. Uptake in monkey brain in a baseline condition contrasted with a blocking condition indicated specific binding of several of the [^11^C]-labelled compounds, of which [^11^C]NOP-1A ((2*S*)-2-[(2-fluorophenyl)methyl]- 3-(2-fluorospiro[4,5-dihydrothieno[2,3-c]pyran- 7,4′-piperidine-1′- yl)-*N*-methyl-propanamide (**36**, Figure 6) was selected for further investigations. In the synthesis developed by Pike et al. [71], [^11^C]NOP-1A (**36**) was prepared from a primary-amide type precursor by methylation with [^11^C]iodomethane in DMSO basified with potassium hydroxide at 80 °C for 5 min. PET imaging experiments with **36** showed a V_T_ in monkey brain ranging from 13 mL cm^−3^ in cerebellum to 21 mL cm^−3^ in amygdala. This fell globally to 8 mL cm^−3^ after blocking with the antagonist SB-612111, indicating a BP_ND_ of 1–2 [132]. A somewhat lower V_T_ range was detected in human brain [133], where the test-retest reliability was about 12% [134].

The NOP/ORL1 antagonist LY2940094 (**37**) exerted a dose-dependent reduction in immobility in the forced swim test, matching that provoked by imipramine, consistent with a potential antidepressant action [23]. Changes in [^11^C]NOP-1A (**36**) binding in brain of living rats revealed the ORL1 occupancy of orally administered LY2940094 (**37**) [135]. MK-0911 (1-(2-fluoroethyl)- 3-[(3*R*,4*R*)-3-(hydroxymethyl)-1-[[(8*S*)-spiro[2.5]octan-8-yl]methyl]piperidin-4-yl]benzimidazol-2- one) is a high affinity, selective NOP receptor antagonist developed by Merck Pharmaceuticals. The fluorine-18 labelled version [^18^F]MK-0911 (**39**) had a V_T_ in human brain ranging from 5 mL cm^−3^ in cerebellum to 7 mL cm^−3^ in temporal cortex, with excellent test-retest stability [72]. Displacement studies with antagonists revealed the presence of a small specific binding component in cerebellum, again raising a red flag about reference tissue quantitation. Studies with the nociceptine ligand [^3^H]PF-7191 (**38**) showed sub-nM binding displacement *in vitro* (*K_i_* = 0.1 nM) and high selectivity over other OR types, as well as promising displaceable binding in rat brain measured *ex vivo* [136].

## 3. Clinical Studies

### 3.1. Age and Gender

The BP_ND_ of [^11^C]Caf (**8**) relative to occipital cortex was 20% lower in thalamus and amygdala of healthy, aged women compared with young women, but tended to increase with age in frontal cortex, whereas increases were more consistently seen in aged men [137]. This finding seems relevant to the age-dependent changes in sensitivity to µOR agonists, compounded by possible gender differences in hepatic tracer metabolism, noted above. Preliminary results with [^11^C]LY2795050 (**30**) PET did not indicate any change in κOR availability with age in humans [138]. Another κ-OR PET study with that ligand showed slightly higher (5–10%) V_T_ in widespread brain regions of male subjects than that seen in age-matched women [139].

### 3.2. Epilepsy

A dual tracer PET study of patients with temporal lobe epilepsy showed increased binding of [^11^C]Caf (**8**) to µORs in the temporal neocortex and decreased binding in the amygdala ipsilateral to the epileptic focus [140]. However, binding of [^11^C]DPN (**15**) to µ- and other OR subtypes did not differ between affected and unaffected cerebral hemipsheres, emphasizing the importance of subtype selectivity in PET studies. Another multi-tracer PET study in temporal lobe epilepsy showed increased µOR binding ([^11^C]Caf (**8**)) in medial inferior temporal cortex and a more widespread increase in δOR binding ([^11^C]MeNTI (**22**)) in the affected temporal lobe [141]. Increased [^11^C]DPN (**15**) binding in temporal pole and fusiform gyrus of epilepsy patients declined with time since last seizure, indicating a transient response of the opioid system [142]. Applying a partial volume correction revealed small post-ictal increases in [^11^C]DPN (**15**) V_T_ in the (sclerotic) hippocampus relative to the interictal state [143], possibly indicating reduced competition from endogenous opioids. Thus, there may be reduced opioid transmission in the post-ictal period.

A study of five patients with reading epilepsy (i.e. seizures provoked by reading text) revealed very circumscribed 10% reductions in [^11^C]DPN binding the temporal parietal cortex in the activation condition compared to non-reading baseline, whereas no such changes was seen in control subjects [144]. In the context of a competition model, the authors interpreted this to indicate task-dependent release of opioid peptides in the patients, but it is difficult to determine the causal relationship between this release and the seizures.

### 3.3. Movement Disorders

In an MPTP model of acquired parkinsonism, a substantial striatal dopamine depletion to FDOPA–PET was associated with a 20–30% reduction in the V_T_ of [^18^F]FcyF, (**11**) in opioid-receptor rich areas, i.e., caudate, anterior putamen, thalamus, and amygdala relative to intact animals [145]. These animals had recovery of their motor function, suggesting that the µOR changes were an adaptive response to dopamine depletion. The same group reported that this effect was (paradoxically) bilateral in animals with unilateral dopamine lesions [146]. In patients with Parkinson’s disease, the [^11^C]DPN (**15**) binding relative to occipital cortex was 20–30% reduced in striatum and thalamus only in those patients with iatrogenic DOPA-dyskinesia, but was unaffected in nondyskinetic Parkinson’s disease patients [147]. This observation seems to merit further investigation, given the disabling effect of dyskinesias often encountered in the treatment of Parkinson’s disease. On the other hand, there was no difference in [^11^C]DPN (**15**) binding in symptomatic DYT1 primary torsion dystonia patients relative to controls [148].

Regional [^11^C]DPN (**15**) binding was unaffected in patients with restless legs syndrome (which, like, Parkinson’s disease, is responsive to levodopa treatment). However, there was a negative correlation between V_T_ and motor symptom severity, and a negative correlation between severity of pain and ligand binding in the medial pain system (medial thalamus, amygdala, caudate nucleus, anterior cingulate gyrus, insular and orbitofrontal cortex). The authors interpreted this result in relation to pain-induced release of endogenous peptides, rather than a primary aspect of restless legs syndrome [149].

Binding of the non-selective OR ligand [^11^C]DPN PET was reduced by 31% in the caudate nucleus and 26% in putamen of a small group of symptomatic Huntington’s disease patients compared to age-matched healthy controls [150]. This effect was less pronounced than was the loss of dopamine transporters seen in the same patients, suggesting that the reduction in ORs may partially accommodate the nigrostriatal degeneration. Despite this finding, there has been no indication for opioid medications in the symptomatic treatment of HD.

### 3.4. Pain

A qualitative study of post pontine infarct central pain showed a reduction in [^11^C]DPN (**15**) uptake in the lateral cerebral cortex on the side contralateral to the main symptoms [151]. A more detailed study indicated [^11^C]DPN (**15**) binding reductions in contralateral thalamus, parietal, secondary somatosensory, insular and lateral prefrontal cortices; these reductions were similar irrespective of the site of the lesion causing the central pain syndrome [152]. Indeed, this network of brain regions is recognized as comprising a central pain pathway. Another [^11^C]DPN (**15**) study of central neuropathic pain showed 15–30% lower OR-binding within the medial pain system (cingulate, insula and thalamus), as well as the inferior parietal cortex of the lateral system (Brodman area 40). Patients with peripheral neuropathic pain had bilateral and symmetrical decreases in [^11^C]DPN (**15**) binding in contrast to the hemispheric changes seen in central pain patients [153]. As always, these binding reductions are ambiguous, perhaps due to reduced receptor expression, increased occupancy, or internalization.

Poor sleep quality in relation to topical application of 10% capsaicin cream (which directly activates cutaneous pain receptors) was associated with higher baseline [^11^C]Caf (**8**) BP_ND_ in the frontal lobe [154]. Thus, we suppose that baseline cortical binding may reflect a tradeoff between pain sensitivity and some cognitive or resilience function subserved by µ-ORs. Capsaicin-induced pain provoked a decrease in [^11^C]Caf (**8**) binding in the contralateral thalamus by as much as 50%, increasing as the subjective severity of the pain [155]. Heat pain reduced the [^18^F]FE-DPN (**16**) binding in limbic and paralimbic brain areas including the rostral ACC and insula [156]. Application of sustained painful stimulus of the jaw muscle with saline injection provoked bilateral reductions in [^11^C]Caf (**8**) binding in the ipsilateral amygdala (5%) and contralateral ventrolateral thalamus (7%) [157]. The same painful stimulus that provoked 5–10% decreases in [^11^C]Caf (**8**) binding in healthy young men tended to increase binding in women; this gender difference was most pronounced in the ventral striatum ipsilateral to the pain [158]. A small group of patients with trigeminal neuralgia had reduced [^11^C]Caf (**8**) binding in left nucleus accumbens, a brain region earlier implicated in pain modulation and response to reward or aversive stimuli [159]. It would be interesting to test if this phenomenon correlated with individual differences in affective state or trait neuroticism.

In another [^11^C]Caf (**8**) study, the reduced binding provoked soon after administration of a sustained pain of moderate intensity had normalized in the hours after cessation of the stimulus [160]. In general, painful stimuli do not desensitize with time, so the relationship between temporal dynamics of opioid signaling and pain perception must be complex. Indeed, pain researchers and clinicians alike are familiar with the phenomenon of allodynia, which is a decrease in pain threshold, or the conversion of previously non-painful stimuli to pain. In a sciatic nerve stimulation model, pain conditioning some hours after stimulation was associated with increased C-fibre response and reduced C-fibre threshold, as well as supraspinal changes marked by increased binding of [^11^C]PEO (**18**) in ipsilateral and bilateral structures of the rat brain [161]. Thus, allodynia may indicate inactivation of pain-mediated opioid release in brain, resulting in greater OR availability. On the other hand, an [^18^F]FE-DPN (**16**) PET study in athletes contrasting receptor availability at rest with the condition immediately after running a half marathon showed reduced binding in various paralimbic and prefontal cortical structures, to an extent correlation with post-running euphoria (“runner’s high”) [162]. Similarly, a [^11^C]Caf (**8**) study in recreationally active men showed a relationship between post-exercising euphoria with decreased µOR binding in widespread cortical areas after high intensity exercise, although effects were less clear after moderate intensity exercise [163].

Prolonged electrical stimulation of the motor cortex for relief of neuropathic pain caused reductions in [^11^C]DPN (**15**) binding in part of the cingulate cortex, prefrontal cortex, the periaqueductal gray prefrontal cortex, and cerebellum [164]. Some of these changes correlated with the extent of pain relief. In a case study, a single session of motor cortex stimulation improved the cold pain threshold, while decreasing [^11^C]Caf (**8**) binding in pain-network brain regions [165]. Other studies showed that low cerebral binding of [^11^C]DPN (**15**) predicted for poor response to motor cortex stimulation for the treatment of neuropathic pain [166]. Stimulation of the central grey for treatment of phantom limb pain provoked a focal decrease in midbrain [^11^C]DPN (**15**) binding, indicating endogenous opioid release [167].

Visceral pain applied by gastric inflammation was without effect on cerebral [^11^C]Caf (**8**) binding in healthy volunteers [168], this standing in contrast to findings with somatic pain as described above. On the other hand, vestibular stimulation provoked decreased [^18^F]FE-DPN (**16**) binding in parts of the right dominant cortical vestibular network [169].

A [^11^C]Caf (**8**) study in which painful heat was applied after administration of supposedly analgesic cream indicated a placebo-mediated reduction in receptor availability [170]. The same group later showed that placebo transcranial direct current stimulation (tDCS) reduced [^11^C]Caf (**8**) BP_ND_ in the periaqueductal gray matter (PAG), precuneus, and thalamus, indicating endogenous opioid release [171]. This placebo effect apparently increased upon administration of verum tDCS. In another study, acupuncture administered according to an authentic analgesic procedure had only slight effects on the binding of [^11^C]DPN (**15**) in human brain [172]. However, a study with [^11^C]Caf (**8**) showed acupuncture therapy to provoke short-term and persistent 10–30% increases in µOR binding in pain-related brain regions; importantly the verum acupuncture condition was contrasted with a sham acupuncture condition in that study [173]. Thus, while acupuncture analgesia may be “in one’s head”, there seems to be a real component mediated by increased opioid transmission. Transcutaneous electrical acupoint stimulation (TEAS) is an analogue of the electrical acupuncture technique. Administration of TEAS at 2 Hz to anesthetized monkey provoked reductions in [^11^C]Caf (**8**) binding in striatum, amygdala and ACC, whereas 100 Hz stimulation had no effect relative to baseline PET recordings [174].

Reminiscent of the findings in the study of pain-induced sleep disturbance noted above, a cross sectional study of sensory processing in healthy volunteers showed lower baseline binding of [^18^F]FE-DPN (**16**) in regions such as insular cortex and orbitofrontal cortex of those with greater sensitivity to cold pain. In addition, there were negative correlations between regional binding and sensory thresholds for non-painful stimuli [175]. Similarly, the individual striatal binding of [^11^C]Caf (**8**) BP_ND_ predicted cold pressor pain threshold, but not cold pressor pain tolerance or tactile sensitivity [176]. A longitudinal [^18^F]FE-DPN (**16**) PET study in neuropathic pain model rats showed lower µ+κ OR availability in the insula, caudate putamen, and motor cortex at three months after the injury [177]. These reductions occurred in association with anhedonia (disinterest in sucrose solution). Overall, these studies suggest that individual differences in OR signaling may mediate vulnerability to environmental stressors, a topic to be elaborated in Section 3.5 below.

Binding of [^11^C]DPN (**15**) was reduced in pineal gland (but not in the brain *per se*) of patients who had been experiencing cluster headache attacks [178], said to be one of the most painful experiences. The authors suggested that inputs from trigeminal nerve to the pineal gland might mediate this change. A group of seven spontaneous migrainers showed ictal reductions in [^11^C]Caf (**8**) binding in the medial prefrontal cortex, which correlated with the baseline or intra-ictal binding [179]. The [^11^C]DPN (**15**) BP_ND_ in brain did not differ between migrainers and healthy controls, nor was there any effect of placebo treatment in either group [180].

### 3.5. Personality, Drug Dependence, and Psychiatric Disorders

Scores in the personality trait of harm avoidance in a group of 23 healthy males correlated positively with binding of [^18^F]FE-DPN (**16**) in the bilateral ventral striatum, suggesting a link with predisposition to substance abuse [181]. It might follow that drug abuse is a kind of self-medication for those with pronounced harm avoidance trait. A comparison of [^11^C]Caf (**8**) uptake in healthy individuals showed that high scores in the harm avoidance trait were associated with high μOR availability in frontal and insular cortex [182], again linking the hard avoidance trait with lower tonic opioid transmission. Score in a scale of behavioral activation, which conceptually guides approach behavior, and notably in a scale designated “fun-seeking”, correlated positively with [^11^C]Caf (**8**) in widespread brain regions [183]. A [^11^C]Caf (**8**) study in 49 healthy volunteers showed an *inverse* relationship between µOR availability in various brain regions and individual scores in the avoidance dimension of interpersonal attachment [184]. Considering the harm avoidance findings, baseline µOR availability may mediate a trade-off between harm avoidance and avoidant behavior in interpersonal relationships, in a psychological analogue of pain or cold sensitivity. 

In a large group of healthy women, [^11^C]Caf (**8**) binding had a negative correlation with BOLD signal responses in amygdala, hippocampus, thalamus, and hypothalamus to viewing emotionally arousing scenes [185]. Non-sexual, albeit pleasurable social touch from a partner provoked widespread increases in [^11^C]Caf (**8**) binding, suggesting reduced opioid signaling [186], whereas social laughter provoked by viewing comedic film clips decreased [^11^C]Caf (**8**) binding in thalamus, caudate nucleus, and anterior insula. Furthermore, baseline µOR availability in some regions was associated with the rate of social laughter [187]. These results are difficult to reconcile, since pleasurable social experiences can seemingly have opposite effects on µOR availability. Contrasting the [^11^C]Caf (**8**) binding in euthymic and unhappy states (provoked by autobiographical reflection) in young women showed higher µOR availability in the rostral anterior cingulate, ventral pallidum, amygdala, and inferior temporal cortex in the unhappy state [188]. This kind of sad reflection provoked greater increases in [^11^C]Caf (**8**) binding in widespread brain regions of women with major depression [189], suggesting an exaggerated opioid response in relation to mood disorder, as distinct from ordinary sadness. A pilot PET study with the κOR-ligand [^11^C]GR103545 (**25**) did not reveal any binding differences between healthy control and patients suffering from major depression [190]. However, a [^11^C]EKAP (**27**) κOR study in healthy volunteers showed an inverse correlation between social status and [^11^C]salvinorin A (**33**) binding in widespread brain areas, with a special association occurring in brain regions mediating reward or aversion [191]. Given the association between social stress and depression, one might have expected covariance κORs in the two studies.

A recent [^11^C]Caf (**8**) PET study of 19 schizophrenia patients and 20 controls showed a 10% lower (Cohen’s d = 0.7) µOR -availability in striatium of the patient group. While such a decrease can hardly be pathogonomic of disease, the authors also reported considerably higher inter-regional covariance of the [^11^C]Caf (**8**) binding in the patients, which might indicate an aberent spatial pattern of opioid signalling in schizophrenia [192]. There have been no OR PET studies in bipolar disorder. 

A [^11^C]Caf (**8**) study showed that circulating levels of the anti-nociceptive cytokine IL-1ra (which correlated with neuroticism scores) predicted for the pain response to a standard stimulus (saline infusion to the masseter muscle), and likewise the reduction in µOR availability in the basal ganglia during the painful stimulus [193]. In a group of female patients suffering from fibromyalgia, [^11^C]Caf (**8**) binding correlated with pain-evoked BOLD signal changes in several brain regions, including dorsolateral prefrontal cortex and nucleus accumbens [194]. Overall, these studies suggest some linking between opioid transmission, mood, and inflammatory markers, which returns to the the notioin that OR signaling may mediate personality traits and vulnerability to stresses of various sorts.

[^11^C]Caf (**8**) PET showed persistently increased µOR binding in frontal and cingulate cortex of acutely detoxified cocaine addicts, which correlated with the extent of craving [195]. Elevated [^11^C]Caf (**8**) binding in frontal and temporal cortical regions was a significant predictor of time to relapse to cocaine use among abstinent addicts [196]. Binge cocaine users showed a significant association between [^11^C]GR103545 (**25**) binding to κORs with the amount of drug consumed. Furthermore, a three-day cocaine binge reduced binding by about 15% [197]. The cerebral binding (V_T_) of the ORL1 ligand [^11^C]NOP-1A (**36**) was globally elevated 10% in detoxified cocaine users [198].

One [^11^C]Caf (**8**) PET study showed persistently increased µOR binding in striatum of detoxified alcoholics, which furthermore correlated with the extent of craving [199]. Abstinent alcoholics showed significantly higher [^11^C]Caf (**8**) binding compared to controls, but a blunting of the response to amphetamine (which indirectly displaces µOR binding), resembling that seen by the same research group in compulsive gamblers [200]. However, others saw only a small increase in [^11^C]DPN (**15**) V_T_ in brain of acutely withdrawn alcoholics, although there was a correlation between individual PET results and craving scores at the time of scanning [201]; the combined (µ+κ) PET signal in that study makes difficult a comparison with [^11^C]Caf (**8**) studies*. Post mortem* autoradiographic examination of [^3^H]DAMGO binding in brain of a large group of alcoholics showed substantial reductions in µOR binding sites, whereas low [^11^C]Caf (**8**) BP_ND_ in ventral striatum of acutely detoxified patients predicted high risk of relapse and poor response to naloxone in interaction with the µOR rs1799971 allele [202]. The disagreement between µOR findings *in vivo* and *post mortem* could indicate low tonic occupancy in alcohol dependent patients, since competition effects would disappear in autoradiographic studies.

In a [^11^C]MeNTI (**22**) PET study, there was globally 10–20% higher δOR binding in brain of a large group of alcohol-dependent subjects; which reached significance upon correcting for age, gender, and smoking status; there was an inverse relationship between binding in some regions and intensity of craving [203]. The V_T_ of the κOR-selective ligand [^11^C]LY2795050 (**30**) was significantly lower in amygdala and pallidum of alcohol-dependent subjects [138]. This stands in contrast to the usual finding of increased µOR binding and the single report of elevated δOR binding.

The naloxone challenge paradigm has a long history in investigations of the regulation of the neuroendocrine axis, but it has been uncertain if naloxone-induced increases in ACTH and cortisol secretion bear any relation to central OR pathways. In a [^11^C]Caf (**8**) PET study of healthy volunteers there were negative relationships between cortisol (but not ACTH) response to naloxone and ligand BP_ND_ in ventral striatum, putamen and caudate [204]. The inverse relationship between naloxone-induced cortisol secretion and [^11^C]Caf (**8**) BP_ND_ in various brain regions of healthy volunteers was absent in alcohol dependent subjects [205]. This suggests that central ORs exert a top-down regulation of the neuroendocrine axis, which might contribute to individual differences in stress response, and that the normal regulation of this axis is disprupted in alcohol dependence. 

There was only a slight difference in [^11^C]Caf (**8**) binding between non-smoking carriers of the µOR rs1799971 allelic variants, but this allelic difference was greater among smokers. Furthermore, the contrast in PET results between active and denicotinized cigarette conditions revealed a positive relationship between reward and altered µOR availability in the smokers [206]. An apparent re-analysis of the same data showed widespread reductions in [^11^C]Caf (**8**) binding after smoking a nicotine-containing cigarette; this effect was moderated by the rs1799971 polymorphism, where carriers of the A allele showed greater response to active cigarette smoking, and higher baseline µOR binding [207]. The authors conceded that non-nicotinergic factors, i.e. conditioning, could be contributing to aspects of smoking related opioid transmission [208].

Smoking subjects with higher dependence, craving, and cigarette consumption rates showed lower baseline [^11^C]Caf (**8**) BP_ND_ in limbic brain regions. There was bluniting of this association during NTX (**5**) treatment [209], but there was very low residual specific binding in the NTX condition, which must have compromised the sensitivity of the method. Another [^11^C]Caf (**8**) PET study showed no difference in BP_ND_ between placebo and active nicotine cigarette conditions, and no difference between smokers and nonsmokers. However, there was a negative correlation in the smokers between BP_ND_ in bilateral superior temporal cortex with scores in an index of nicotine dependence [210].

As noted above, challenge with amphetamine can indirectly provoke increased opioid peptide release. However, in a placebo-controlled, double-blinded and randomized [^11^C]Caf (**8**) PET study, amphetamine challenge (0.3 mg/kg) did not alter µOR availability in healthy male volunteers [211]. This stands in contrast to another study, wherein a high dose of amphetamine (0.5 mg/kg) provoked reductions in [^11^C]Caf (**8**) binding in widespread brain regions, i.e. frontal cortex, putamen, caudate, thalamus, anterior cingulate, and insula, whereas a sub-pharmacological dose was without such an effect [212,213]. Preclinical studies point to the importance of receptor internalization on the vulnerability of OR-receptor binding to challenge with amphetamine [214]. Notwithstanding this caveat, amphetamine induced reductions in [^11^C]Caf (**8**) binding were blunted in compulsive gamblers compared to that in a healthy control group controlled for smoking and drinking [215]. There was a general correlation between dopamine synthesis capacity to FDOPA PET and [^11^C]Caf (**8**) binding in putamen of healthy controls, and likewise in pathological gamblers, consistent with a tight relationship between dopamine and opioid systems in relation to compulsive behaviors [216].

Women with bulimia nervosa had reduced [^11^C]Caf (**8**) binding in the left insula, to an extent correlating with their duration of fasting [217]. Obese patients (BMI 40) had globally 20% lower [^11^C]Caf (**8**) BP_ND_ compared to lean volunteers; contrary to some reports, the same obese subjects had normal dopamine D_2_ receptor levels in striatum [218]. There were similar reductions in [^11^C]Caf (**8**) binding in morbidly obese subjects and patients with binge eating disorder [219]. Weight loss after bariatric surgery for the treatment of obesity resulted in a global 25% increase of µOR binding [220]. A dual tracer study with [^11^C]Caf (**8**) and the dopamine receptor ligand [^11^C]raclopride showed a high correlation in the striatum of lean subjects, whereas this correlation was considerably weaker in the ventral (limbic) striatum of the morbidly obese, suggesting an uncoupling of opioid/dopamine interactions in that condition [221]. This finding might predict analogous results in gambling and substance abuse disorders, which likewise may involve dysregulation opioid/dopamine interactions.

Feeding, regardless of the hedonic experience (palatable versus unpalatable meal), provoked widespread decreases in [^11^C]Caf (**8**) binding in non-obese healthy subjects, suggesting that OR transmission mediates some aspect of the rewarding properties of food [222]. Also in non-obese subjects, [^11^C]Caf (**8**) BP_ND_ in amygdala correlated inversely with BMI in the range 20–27 [223]. In that study, BP_ND_ in other brain regions correlated with the BOLD signal response in orbitofrontal cortex upon viewing palatable food. In a group of lean subjects, the [^11^C]Caf (**8**) BP_ND_ at baseline in widespread brain regions correlated with BOLD responses to viewing palatable food [224], suggesting that low basal occupancy increases the response to cues. Interestingly, exercise increased or decreased thalamic µOR binding in these subjects; the direction of this change correlated with the individual BOLD signal in the contrast between viewing palatable and non-palatable food. This draws attention to individual differences in effects of exercise on the hedonic response to food, which may have some bearing on the relationship between exercise and weight loss, with the caveat that only intense exercise may significantly engage opioid transmission, as claimed above. Other studies show widespread reductions in µOR availability in frontolimbic regions after high intensity aerobic exercise, in correlation with negative affect. In contrast, mean binding was unaltered after moderate-intensity exercise, although there was some positive association with euphoria [225]; too much of a good thing spoils runner’s high, it seems.

## 4. Conclusions and Outlook

The past decades have seen extraordinary progress in the development of ligands for PET studies of ORs. Early radiopharmaceutical research focused on studies with the antagonist [^11^C]DPN (**15**) and the µOR-selective agonist [^11^C]Caf (**8**), and the great preponderance of human PET studies have employed these and closely related tracers. While studies with non-selective tracers reveal the composite of OR binding sites, specific tracers may be more indicative of physiological changes in disease states. Attaining high molar activity is of great importance in PET studies with [^11^C]Caf (**8**) and other highly potent agonist ligands; fortunately, most tracers described in this review have molar activities of at least 50 GBq/µmol, corresponding to an injected mass of about 1 µg of the drug. This is hardly a relevant dose in the case of antagonist ligands, and would give a comfortable 100-fold margin of safety with the potent µOR agonist [^11^C]Caf (**8**). 

The µOR ligands have the useful property of binding in competition with endogenous opioid peptides, such that changes in the uptake in PET studies can reveal altered endogenous opioid release under various physiological conditions. This model has been particularly useful in studies of pain pathways, which largely involve µORs in telencephalon, and in some pharmacological or behavioral activation studies. However, the simple competition model may be inadequate to account for all observations. Thus, one of the [^11^C]Caf (**8**) studies noted above reports widespread reductions in µOR availability after smoking [207], despite the 30–3000-fold lower affinity of endogenous opioid peptides at µOR. This would seem to imply an implausibly enormous increase in peptide release to effect such changes by competition alone.

Pain studies have so far dominated the field of clinical PET research with OR-ligands, with relatively few reports on other models or conditions, as summarized in Table 2. For example, there in only one PET study of opioid receptors in schizophrenia, and only scant documentation in depression, or for that matter, in a range of common neurological disorders. In several human diseases noted above, the OR binding may be only 10% higher or lower than in the control group; while these small differences can have a large effect size, it is perhaps difficult to argue that such small differences can be causative of complex disorders or symptoms. 

Addiction research using PET studies of ORs are so far mostly confined to alcohol, cocaine, and nicotine abuse and (strangely, perhaps), opioid addiction has hardly been a research theme, other than in a few occupancy studies. Since antagonists are relatively safe at doses provoking high occupancy (*viz* 50 mg naloxone for opioid overdose), we suppose that the B_max_ of ORs might be determinable in relation to opioid addiction and withdrawal by conducting serial PET studies over a range of molar activity, even in the presence of significant agonist occupancy. Indeed, chronic morphine was reported 45 years ago to increase the abundance of [^3^H]naloxone binding sites in rat brain [226], but no such studies are reported in human opioid users, despite the catastrophe of the current opioid addiction epidemic. This kind of information might help to understand better the correlates of addiction and relapse. In addition, genetic studies of dopaminergic and opioid systems in relation to addiction [227], in conjunction with molecular imaging studies, could help to establish better the risk factors for opioid addiction. Endomorphins and other novel opioid petpides may present new avenues for obtaining opioid analgesia [228], while moderating the risk of “iatrogenic opioid addiction”. The development of PET tracers with good binding properties *in vivo* and high selectivity for ORs other than the µ-type has accelerated in the past decade. However, there remain relatively few clinical molecular imaging studies of these important targets. Thus, developments in radioligand chemistry have for the presence to read for the present presence outpaced clinical PET imaging, a state of affairs that could enable and motivate a broad range of studies focusing on non-µORs over the coming decades. Just for example, κORs have an established role in the reinstatement of stress induced drug use in experimental animals, i.e. nicotine use [229], and very recent results indicate a relationship between κORs and stress-induced binge cocaine use [197].

## Figures and Tables

**Figure 1 molecules-24-04190-f001:**
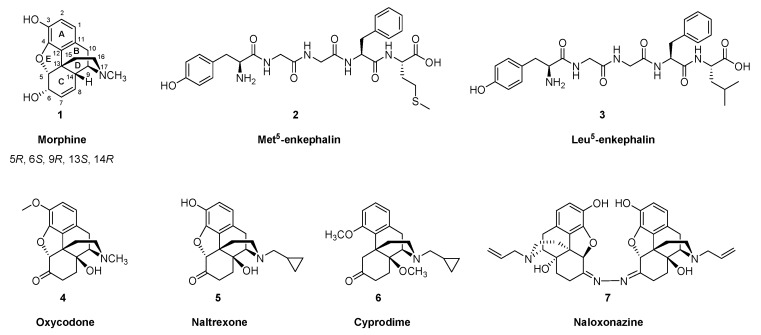
Chemical structures of endogenous opioid peptides and selected opioid receptor ligands.

**Figure 2 molecules-24-04190-f002:**
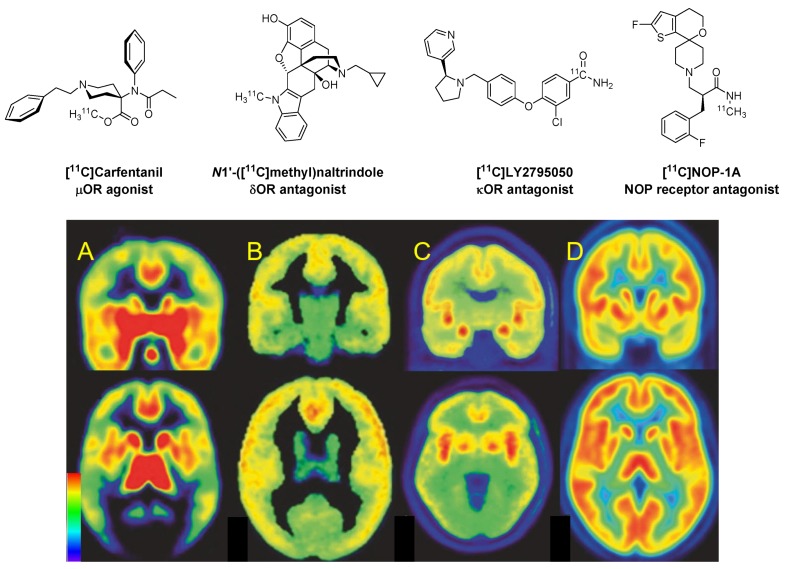
Human brain opioid receptor PET images in coronal (upper row) and axial (lower row) planes. Modified and reproduced with permission from Peciña et al. [19]. From left to right, we see (**A**) the µOR agonist [^11^C]carfentanil, binding most abundantly in the caudate nucleus, anterior cingulate cortex, thalamus, and pituitary gland; (**B**) the δOR antagonist *N*1′-([^11^C]methyl)natrindole, which has diffuse binding throughout neocortex; (**C**) the κOR antagonist [^11^C]LY2795050, which has high binding in the insular cortex, lateral frontal cortex and amygdala; (**D**) the NOP antagonist [^11^C]NOP-1A, which binds abundantly throughout the brain. Binding sites of µ-, κ- and NOP-OR ligands are expressed as binding potential relative to the cerebellum (BP_ND_), whereas binding of the δ-ligand (which has no non-binding reference region) is expressed as net influx (*K_i_*,) in units of perfusion (mL cm^−3^ min^−1^). The color scale in the lower right indicates (for **A**, **C**, and **D**) BP_ND_ ranging from 0 to 2, or (**B**) *K_i_* ranging from 0–0.1 mL cm^−3^ min^−1^

**Figure 3 molecules-24-04190-f003:**
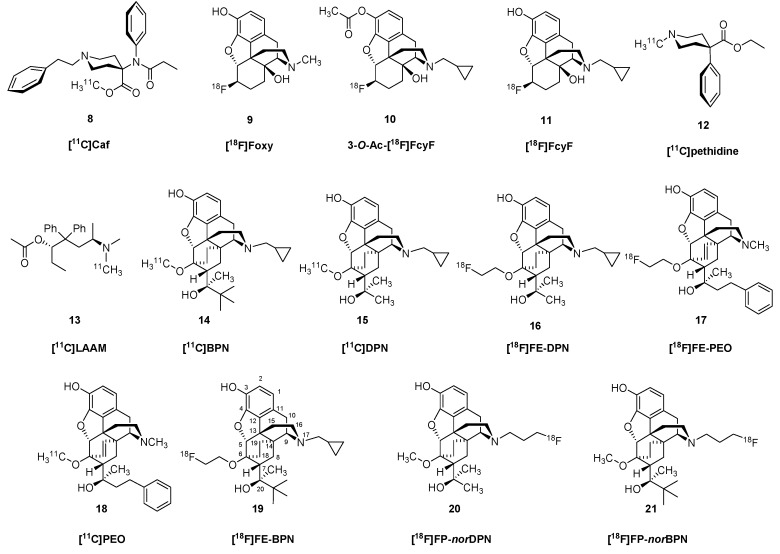
Structures of µ-selective and non-selective opioid receptor radioligands.

**Figure 4 molecules-24-04190-f004:**
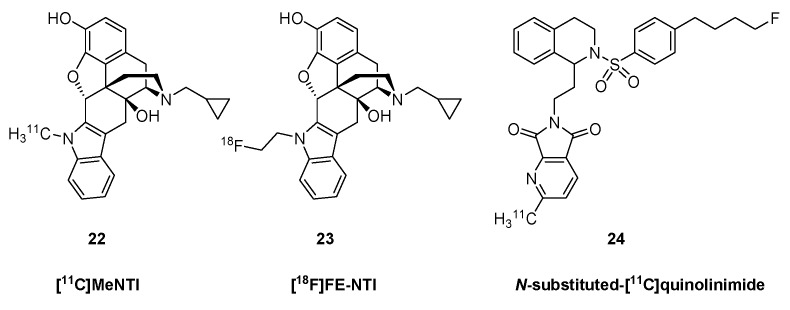
Labeled δ-opioid receptor ligands.

**Figure 5 molecules-24-04190-f005:**
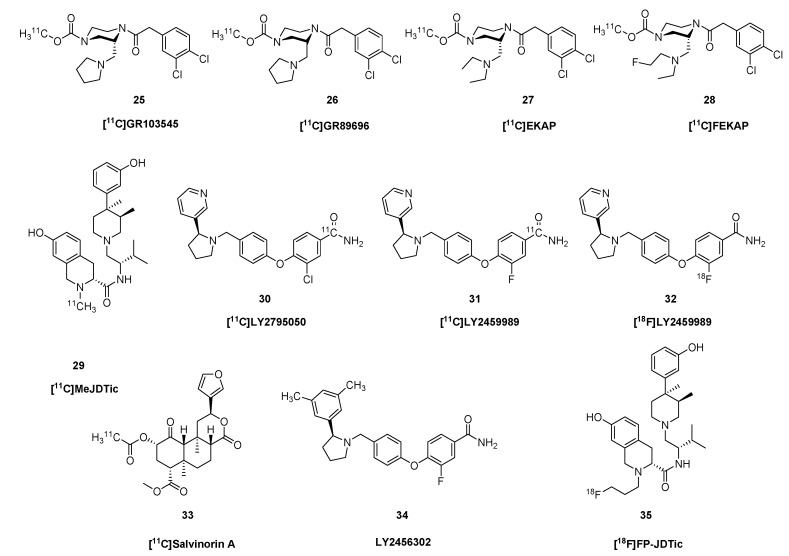
Selected κOR ligands.

**Figure 6 molecules-24-04190-f006:**
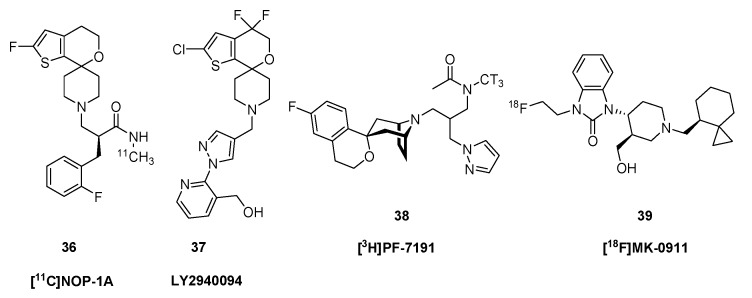
Chemical structures of selected ORL1 receptor ligands.

**Table 1 molecules-24-04190-t001:** Binding profile of selected ligands at the human opioid receptors [51] ^.^

Ligand	Ki [nM]				Action	Compound Class	Ref.
	μ-OR	δ-OR	κ-OR	NOP			
Met^5^-enkephalin	261	9.9	-	-	agonist δOR	EOP	Stefanucci [52]
Leu^5^-enkephalin	513	10.7	-	-	agonist δOR	EOP	Stefanucci [52]
β-Endorphin	2.1	2.4	96	-	agonist μOR, κOR	EOP	Corbett [53]
Dinorphin A	1.6	1.25	0.05	386	agonist κOR	EOP	Zhang [54]
Nociceptin	437	2846	147	0.08	agonist	EOP	Cami-Cobeci [50]
Morphine	2.06	>10,000	134	>10,000	agonist μOR	EM	Valenzano [55]
Oxycodone	16	7680	43,000	-	agonist μOR	EM	Miyazaki [56]
Naltrexone	0.62	12.3	1.88	-	antagonist	EM	Zheng [57]
Carfentanil	0.07	-	-	-	agonist μOR	4-AP	Henriksen [58]
Carfentanil^a^	0.051	4.7	13	-	agonist μOR	4-AP	Frost [59]
Carfentanil^b^	0.024	3.28	43.1	-	agonist μOR	4-AP	Cometta [60]
Cyclofoxy^a^	2.62	89	9.3	-	antagonist μOR, κOR	EM	Rothman [38]
DPN	0.07	0.23	0.02	-	antagonist	orvinol	Raynor [61]
BPN	1.5	6.1	2.5	77.4	partial μOR agonist, κOR antagonist	orvinol	Cami-Cobeci [50]
PEO	0.18	5.1	0.12	-	full agonist	orvinol	Marton [62]
FE-DPN	0.24	8.00	0.20	-	antagonist	orvinol	Schoultz [63]
FE-BPN	0.24	2.10	0.12	-	mixed agonist/antagonist	orvinol	Schoultz [63]
FE-PEO	0.10	0.49	0.08	-	full agonist	orvinol	Schoultz [63]
NTI^b^	3.8	0.03	332	-	antagonist δOR	EM	Portoghese [64]
MeNTI^b^	14	0.02	65	-	antagonist δOR	EM	Portoghese [64]
GR103545	16.2	536	0.02	-	agonist κOR	ArAP	Schoultz [65]
LY2459989^a^	7.68	91.3	0.18	-	antagonist κOR	APPB	Zheng [66]
LY2795050	25.8	153	0.72	-	antagonist κOR	APPB	Zheng [57]
FEKAP	7.4	139	0.43	-	agonist κOR	ArAP	Li [67]
EKAP	8.6	386	0.28	-	agonist	ArAP	Li [68]
MeJDTic	8.88	118	1.01	-	antagonist κOR	JDTic	Poisnel [69]
Salvinorin A	>1000	5790	1.9	-	agonist κOR	NND	Harding [70]
NOP-1A	-	-	-	0.15	antagonist NOP	FDPTP	Pike [71]
MK-0911	94	-	-	0.6	antagonist NOP	SPB	Hostetler [72]

EOP: Endogenous opioid peptide, EM: 4,5-Eopxy-morphinan, 4-AP: 4-Anilidopiperidine, orvinol: 6,14-ethenomorphinan, Bentley-compound, ArAP: Arylacetamidopiperazine, APPB: Aryl- phenylpyrrolidinylmethyl-phenoxy-benzamide, JDTic: *trans*-3,4-dimethyl-4-(3-hydroxyphenyl)- piperidine, **NND**: “non nitrogenous” diterpene, FDPTP: 2′-fluoro-4′,5′-dihydrospiro[piperidine- 4,7′-thieno [2,3-c]pyran]- derivative, SPB: [[spiro[2.5]octan-8-yl]-methyl]piperidin-4-yl] benzimidazol-2-one, ^a^: in the rat brain ^b^: in guinea pig brain membranes.

**Table 2 molecules-24-04190-t002:** A summary of the key results with opioid PET in clinical research.

Condition	Ligand	Main Finding	Ref.
Healthy aging	[^11^C]Caf (**8**) (µOR)	20% decrease in frontal cortex (females)	[137]
Epilepsy	[^11^C]Caf (**8**) (µOR)	Increased in ipsilateral temporal lobe, decreased in amygdala	[140]
Epilepsy	[^11^C]DPN (**15**) (mixed ligand)	No change	[140]
Parkinson’s disease	[^11^C]DPN (**15**) (mixed ligand)	20–30% decrease in striatum and thalamus only in those patients with iatrogenic DOPA-dyskinesia	[147]
Huntington’s disease	[^11^C]DPN (**15**) (mixed ligand)	30% reduced in caudate/putamen	[150]
Pontine infarct central pain	[^11^C]DPN (**15**) (mixed ligand)	Reduced throughout pain network	[151]
Capsaicin-induced acute pain	[^11^C]Caf (**8**) (µOR)	Up to 50% decrease contralateral thalamus, in proportion to subjective severity	[155]
Sustained painful stimulus of the jaw muscle with saline injection	[^11^C]Caf (**8**) (µOR)	Blateral decrease in binding in the ipsilateral amygdala (5%) and contralateral ventro-lateral thalamus (7%)	[157]
painful heat	[^11^C]Caf (**8**) (µOR)	Placebo effect on binding changes	[170]
Acupuncture therapy with sham acupuncture control	[^11^C]Caf (**8**) (µOR)	Persistent 10–30% increases in µ-OR binding in pain-related brain regions	[173]
Harm avoidance trait in healthy males	[^18^F]FE-DPN (**16**) (mixed ligand)	Trait correlated positively with binding in vental striatum, suggesting link with substance abuse	[181]
Correlation with BOLD signal responses A large group of healthy women, binding. to viewing emotionally arousing scenes	[^11^C]Caf (**8**) (µOR)	Negative correlation in amygdala, hippocampus, thalamus, and hypothalamus	[185]
Major depressive disorder	[^11^C]GR103545 (**25**) (κOR)	No difference from controls	[190]
Detoxified cocaine addicts	[^11^C]Caf (**8**) (µOR)	Increased in frontal and cingulate cortex, which correlated with the extent of craving	[195]
Detoxified alcohol-dependent subjects	[^11^C]MeNTI (**22**) (δOR)	Globally 10–20% increased binding inverse relationship in some regions with intensity of craving	[203]
Obesity (BMI > 40)	[^11^C]Caf (**8**) (µOR)	Globally 20% lower compared to lean volunteers	[218]
Feeding, regardless of the hedonic experience in non-obese subjects	[^11^C]Caf (**8**) (µOR)	Widespread decreases in binding	[222]

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
