# Peer review of "A Survey of Molecular Imaging of Opioid Receptors"

_molecules, 2019, doi:10.3390/molecules24224190_

Round 1
Reviewer 1 Report
The manuscript reviews significant PET radioligands for opioid receptors, with focus on those having major clinical impacts. Overall such review could provide good educational value. It is particularly useful that the review is structured in the orders of radiochemistry, brief pre-clinical description and clinical use. The summary tables of abbreviation, basic properties and clinical research are also helpful. The manuscript needs revision to harmonize the styles, terms written by different contributors and to correct typos.
Specifically, please make the following changes:
1. Revise the statement in Introduction, lines 56/57 that says “Since opioid molecular imaging has not had a systematic review in recent years”. In fact, Dannals (2013 in Journal of Labelled Compounds and Radiopharmaceuticals) and Henricksen (2008 in Brain) published good reviews on the topic. These two papers should be cited.
2. As said earlier, one of the useful features of the manuscript is to describe radiochemistry followed by pre-clinical studies. There are a few places this feature is not harmonized and somewhat creates confusion. For example, the paragraph in lines 215-228 discusses both [11C]BPN and [11C]DPN before the synthesis of [11C]DPN is described. Those sentences on [11C]DPN could be reorganized to where [11C]DPN is discussed. The same issue needs to be addressed for the paragraph in lines 426-434. A sudden appearance of discussion on [11C]GR103545 following the synthesis of [11C]LY2459989 is not appropriate.
3. Consistently cite the compound number immediately after the compound name. For example line 416 for 31, line 423 for 32, line 444 for 33 and many other places.
4. Chemical structures in Fig 4, Fig 5 should have the same % reduction in size as in other figures.
5. Please use the recommended (see Innis et al, Journal of Cerebral Blood Flow & Metabolism (2007) 27, 1533–1539) unit of mL cm-3 for all VT values.
6. Correct typos such as:
a. Delete “-“ in radiotracer names. For example, [11C]LY2459989, not [11C]-LY2459989.
b. Lines 37-38, use “;” to separate keywords.
c. Line 94, use recommended unit for K1 (not Ki).
d. Lines 160/165, use the same name for 10.
e. Line 195-196, delete duplicate sentence “Whereas BPN is a mixed agonist-antagonist with high affinity for μ- and κORs”.
f. Lines 315, 339, 374, always use [11C]iodomethane.
g. Line 449, salvinorin not salvinorum.
h. Line 457, add °C after 150.
i. Line 498, secondary-amide, not secondary-amine.
j. Line 512, [18F]MK-0911 is 38, not 31. Also not [18F]-MK0911.
k. Line 558, “defect” for “difference”?
l. Line 569, “than” for “that”.
m. Line 693, “bipolar”.
n. Line 702, “[11C]Caf (8) PET”, not “[11C]Caf PET (8)”.
o. Line 789, “4. Conclusions and outlook”, not “5.”.
Author Response
Revise the statement in Introduction, lines 56/57 that says “Since opioid molecular imaging has not had a systematic review in recent years”. In fact, Dannals (2013 in Journal of Labelled Compounds and Radiopharmaceuticals) and Henricksen (2008 in Brain) published good reviews on the topic. These two papers should be cited. As said earlier, one of the useful features of the manuscript is to describe radiochemistry followed by pre-clinical studies. There are a few places this feature is not harmonized and somewhat creates confusion. For example, the paragraph in lines 215-228 discusses both [11C]BPN and [11C]DPN before the synthesis of [11C]DPN is described. Those sentences on [11C]DPN could be reorganized to where [11C]DPN is discussed. DoneThe same issue needs to be addressed for the paragraph in lines 426-434. A sudden appearance of discussion on [11C]GR103545 following the synthesis of [11C]LY2459989 is not appropriate. Corrected
Consistently cite the compound number immediately after the compound name. For example line 416 for 31, line 423 for 32, line 444 for 33 and many other places. Nomenclature and numbering are now applied consistently Chemical structures in Fig 4, Fig 5 should have the same % reduction in size as in other figures. Corrected Please use the recommended (see Innis et al, Journal of Cerebral Blood Flow & Metabolism (2007) 27, 1533–1539) unit of mL cm-3 for all VT values. Done Correct typos such as: Delete “-“ in radiotracer names. For example, [11C]LY2459989, not [11C]-LY2459989. Done Lines 37-38, use “;” to separate keywords. Done Line 94, use recommended unit for K1 (not Ki). Done. Note that K1 and Ki have the same units, mL cm-3 min-1 Lines 160/165, use the same name for 10. Corrected . Line 195-196, delete duplicate sentence “Whereas BPN is a mixed agonist-antagonist with high affinity for μ- and κORs”. Corrected Lines 315, 339, 374, always use [11C]iodomethane. Globally corrected Line 449, salvinorin not salvinorum. Corrected Line 457, add °C after 150 Corrected. Line 498, secondary-amide, not secondary-amine. Corrected Line 512, [18F]MK-0911 is 38, not 31. Also not [18F]-MK0911. All corrected. Line 558, “defect” for “difference”? Fixed Line 569, “than” for “that”. Fixed Line 693, “bipolar”. Corrected Line 702, “[11C]Caf (8) PET”, not “[11C]Caf PET (8)”. Corrected Line 789, “4. Conclusions and outlook”, not “5.”. Corrected
Reviewer 2 Report
The review is well written it is easily readable.
This reviewer considers that it is important to conclude about the perspectives on new approaches and the promising or potential translational approaches. Also, it is important to consider the new challenges on imaging and/or treatment with endogenous opioids.
Author Response
Q: This reviewer considers that it is important to conclude about the perspectives on new approaches and the promising or potential translational approaches. Also, it is important to consider the new challenges on imaging and/or treatment with endogenous opioids.
A: As we have already suggested in the conclusion, there is a paucity of studies using selective ligands for several of the OR subtypes, although there has been considerable progress in radiochemistry development. So, it is time for clinical PET research to make proper use of the new tools that have become available. We also point to the need for OR-PET research in the context of addiction research, despite the logistic difficulties therein. The conclusion has been revised to emphasize better these points. The final paragraph now reads more like this:
Indeed, chronic morphine was reported 45 years ago to increase the abundance of [3H]naloxone binding sites in rat brain (PMID: 938584), but no such studies are reported in human opioid users; such studies might help to understand better the correlates of addiction and relapse. In addition, genetic studies of dopaminergic and opioid systems in relation to addiction (PMID 31620026), in conjunction with molecular imaging studies, could help to establish better the risk factors for opioid addiction. Endomorphins and other novel opioid petpides may present new avenues for obtaining opioid analgesia (PMID 29132133), while moderating the risk of “iatrogenic opioid addiction”.
Q: The development of PET tracers with good binding properties in vivo and high selectivity for ORs other than the µ-type has accelerated in the past decade. However, there remain relatively few clinical molecular imaging studies of these important targets.
A: Thus, developments in radioligand chemistry have outpaced clinical PET imaging, as state of affairs that motivate a broad range of studies focusing on non-µORs over the coming decades. Just for example, κORs have an established role in the reinstatement of stress induced drug use in experimental animals, i.e. nicotine use (PMID 18575850), and very recent results indicate a relationship between κORs and stress-induced binge cocaine use (PMID 31026862).
Reviewer 3 Report
This is a very timely article considering the overwhelming opioid addiction crisis. The article is well written and comprehensive. The authors are to be congratulated on pulling this information together.
I have a few suggestions and minor corrections that authors might consider to make the reading easier. Also a few places that require attention and change.
My major concern is the title. It deals entirely with PET agents and I believe this should be in the title. I would suggest expanding the title to: A survey of PET radiopharmaceuticals for molecular imaging of Opioid Receptors.
Below are suggested changes in the order that they appear in the manuscript:
Line 29 remove the and after Caf
Line 122 remove the of between density of in
Line 153 No reference to to the subject of the sentence starting In contrast, ....
Lines 159-160 The list contains 4 different versions of 10 - some separated by commas. I would suggest we treat this as a list and use semi colons to separate the different versions. This problem also occurs in a few other places.
Line 163 FcyF should be CyF (it is none radioactive!). (see table of abbreviations Page 20)
Line 195 Delete Whereas that starts the sentence -not required.
Line 225 The paragraph starting on line 225 ending 228 should be referenced.
Line 290 Insert is between toxicity and a
Line 312 add nM after the = 0.12 at the end of this line
Line 332 In this line and many other places ml is used. You may want to consider mL . If you make this decision please change all occurrences using Find.
Line 380 Hopefully this line will transfer to a new page - it being a heading
Lines 400-401 This sentence describes the wrong compound. It describes LY-2795050 not GR103545! This needs to be changed.
Lines 416-420 The preparation of [11C]LY2459989 (31) is not referenced. This should quote Zheng et al., J Nucl. Med. 2014:55: 1185-1191 that you have used later Reference 113
Line 435 Replace natural with naturally
Line 453 Insert is between that ring
Line 512 (31) should read (38)
Line 548 Insert a between In MPTP
Line 569 Replace that with than
Line 620 [11C]DPN is incorrect - replace with [11C]carfenanil
Line 680 Reference 173 is totally incorrect. Please find correct reference
Line 740 Insert in between contrast PET
Line 1257 No Journal title!
Line 1363 This reference is incorrect
Author Response
Q: This is a very timely article considering the overwhelming opioid addiction crisis. The article is well written and comprehensive. The authors are to be congratulated on pulling this information together.
I have a few suggestions and minor corrections that authors might consider to make the reading easier. Also a few places that require attention and change. My major concern is the title. It deals entirely with PET agents and I believe this should be in the title. I would suggest expanding the title to: A survey of PET radiopharmaceuticals for molecular imaging of Opioid Receptors.
A: We chose to omit reference to SPECT ligands for molecular imaging. So it is probably better to leave the title as it is currently expressed, given the nearly complete emphasis on PET ligands.
Line 29 remove the and after Caf Corrected
Line 122 remove the of between density of in Corrected
Line 153 No reference to to the subject of the sentence starting In contrast, . Corrected at present line 155. ..
Lines 159-160 The list contains 4 different versions of 10 - some separated by commas. I would suggest we treat this as a list and use semi colons to separate the different versions. This problem also occurs in a few other places. Corrected
Line 163 FcyF should be CyF (it is none radioactive!). (see table of abbreviations Page 20) Corrected
Line 195 Delete Whereas that starts the sentence -not required. We prefer the leave this in, to emphasis the contrasting agonist/antagonist properties.
Line 225 The paragraph starting on line 225 ending 228 should be referenced Corrected
Line 290 Insert is between toxicity and a Corrected
Line 312 add nM after the = 0.12 at the end of this line Corrected
Line 332 In this line and many other places ml is used. You may want to consider mL . If you make this decision please change all occurrences using Find. Globally corrected to mL
Line 380 Hopefully this line will transfer to a new page - it being a heading. Of course, there will be a line-break here.
Lines 400-401 This sentence describes the wrong compound. It describes LY-2795050 not GR103545! This needs to be changed. The incorrect text is deleted.
Lines 416-420 The preparation of [11C]LY2459989 (31) is not referenced. This should quote Zheng et al., J Nucl. Med. 2014:55: 1185-1191 that you have used later Reference 113 This is now correctly cited in the present line 428.
Line 435 Replace natural with naturally Corrected
Line 453 Insert is between that ring Corrected
Line 512 (31) should read (38) Corrected
Line 548 Insert a between In MPTP Corrected
Line 569 Replace that with than Corrected
Line 620 [11C]DPN is incorrect - replace with [11C]carfenanil Corrected
Line 680 Reference 173 is totally incorrect. Please find correct reference Corrected to Prossin AR, Koch AE, Campbell PL et al. PMID: 26283642
Line 740 Insert in between contrast PET Correct
Line 1257 No Journal title! Journal title AJNR Am J Neuroradiol is added
Line 1363 This reference is incorrect Fixed as in ref. in former line 680